# Association between physical measures of spinopelvic alignment and physical functioning with patient reported outcome measures after total hip arthroplasty: Systematic review and narrative synthesis

Sima Vatandoost[1]*, Saghar Soltanabadi[1], Renee-Marie Ragguett[2], Lucas Fernandino[3], Brent Lanting[4], K.C. Geoffrey Ng[5], Katie Kowalski[1], Alison Rushton[1]

1 School of Physical Therapy, Western University, London, Ontario, Canada, 2 Schulich School of Medicine and Dentistry, Western University, London, Ontario, Canada, 3 Department of Basic and Health Sciences, Centro Universitario de Belo Horizonte, Belo Horizonte, Minas Gerais, Brazil, 4 Department of Surgery, Division of Orthopaedic Surgery, Schulich School of Medicine and Dentistry, Western University, London, Ontario, Canada, 5 Department of Medical Biophysics, Schulich School of Medicine and Dentistry, Western University, London, Ontario, Canada

* svatand@uwo.ca

## Abstract

### Background

The prevalence of total hip arthroplasty (THA) is increasing, making optimizing post-surgery outcomes essential. Co-occurrence of low back pain (LBP) is frequently reported. Patient-reported outcome measures (PROMs) are commonly used to evaluate outcomes but have limitations, and physical measures may have value. This study evaluated association between 1] physical measures of spinopelvic alignment (measuring from spine to the pelvis) and physical functioning with PROMs after THA, and 2] physical measures with LBP.

### Methods

Systematic review (PROSPERO: CRD42023412744) searched electronic databases, grey literature, and key journals to 31/5/2024 for cross-sectional and longitudinal studies assessing associations following THA. Two independent reviewers evaluated eligibility, extracted data, and assessed risk of bias (RoB, NIH tool). Owing to heterogeneity, a narrative synthesis was conducted, and GRADE determined overall quality.

### Results

Fifty-one studies were included: 8 low-RoB, 24 moderate-RoB, 19 high-RoB. The evidence on spinopelvic alignment was extremely limited (6 studies), with the main synthesis focusing predominantly on physical measures of physical functioning.

**Data availability statement:** All relevant data are within the manuscript and its Supporting Information files, and our submission contains all raw data required to replicate the results of our study.

**Funding:** The author(s) received no specific funding for this work.

**Competing interests:** The authors have declared that no competing interests exist.

Very low-quality evidence supports no cross-sectional association between walking speed with function in short-term, and between light-intensity physical activity with patient-reported physical activity in medium-term. Very low-quality evidence supports no longitudinal association between Timed-Up-and-Go change with pain/function change, and Stair-Climb-Test (SCT) change with pain/function change in short-term, and SCT change with function change in medium-term. Other findings were inconsistent. Three LBP studies could not be synthesized due to heterogeneity.

## Conclusions

Very low-quality evidence supports no association between physical measures of physical functioning with PROMs, with limited LBP studies. There was significant heterogeneity, with few low ROB studies, highlighting a need for future research on the association between the measures.

## Introduction

Total hip arthroplasty (THA) is successful in reducing pain, restoring function and Quality of Life (QoL), with good long-term outcomes in patients with end-stage hip osteoarthritis (OA) [1,2]. The number of procedures is projected to increase to 659% by 2060 [3] due to factors such as aging population and obesity [4,5]. Optimizing outcomes following THA is essential to reduce complications and revisions [6] and improve function and QoL. Patient-centered care places an emphasis on patient-reported outcome measures (PROMs) to evaluate outcomes [7], but they have some limitations, e.g., ceiling effects [8]. A lack of correlation with physical measures [9,10] suggests both types of measures could provide a comprehensive assessment [11].

For THA, physical measures of spinopelvic alignment and physical functioning are important [12,13]. Spinopelvic alignment is the relationship between the spine and pelvic regions [14]. Severe hip OA can lead to abnormal spinal alignment (e.g., a hip flexion contracture that increases anterior pelvic tilt and lumbar lordosis) [15], often linked to poor clinical outcomes, e.g., low back pain (LBP) [16]. It has been found that preoperative spinopelvic parameters that lead to altered mechanics can be normalized one year following THA [17], with reduced lumbar scoliosis and improved LBP [18]. However, findings from other studies were contradictory. Ben-Galim et al. [16] found LBP improved following THA without changes in spinopelvic parameters, but Eyvazov et al. demonstrated that changes in spinopelvic alignment were not related to LBP improvement [19]. In a recent scoping review that provided an overview of spinopelvic alignment and LBP following THA, inconsistent findings and methodological issues of the studies were reported [20]. Although interest in spinopelvic alignment and THA outcomes is growing, systematic reviews focusing specifically on physical measures linking spinal and pelvic reference points remain limited.

As recommended by Osteoarthritis Research Society International (OARSI), physical measures that evaluate activities of daily living are complementary to PROMs for patients with hip OA [11]. However, findings are again inconsistent. While some

studies reported an association between physical performance-based outcome measures and PROMs for function following THA [21–23], other studies found no/weak relationship [9,24,25]. Previous systematic reviews showed an association between preoperative function or pain with clinical or functional outcomes following THA. However, these studies examined outcomes measured with either PROMs or physical measures [26–30]. Therefore, little is known about the association between physical measures of spinopelvic alignment and physical functioning with PROMs following THA.

### Objectives

The primary objective of this systematic review was to evaluate the association between physical measures of spinopelvic alignment and physical functioning with PROMs following THA. The secondary objective was to evaluate the association between the physical measures with LBP following THA.

## Materials and methods

### Design

This systematic review was written following a predefined published [31] and PROSPERO registered protocol (CRD42023412744). All steps in the registered protocol were followed, with any deviations reported. Due to limited studies retrieved in an initial scoping search, no restrictions were first placed on THA indications, resulting in a broad population. To reduce heterogeneity, the population was restricted to THA participants for hip OA excluding secondary hip OA. Clinician-based questionnaires were also excluded as they may not accurately evaluate the functional status [32]. The review search was originally planned until July 31, 2023. Because of significant heterogeneity hindering data synthesis, the search date was extended. Based on Rushton et al.'s review [33] of pain and disability following primary lumbar discectomy, medium-term follow-up were initially defined as >3, ≤ 12 months. Given no significant difference in outcomes between 1 and 2 years post-THA [34], it was revised to >3 months to ≤2 years, with long-term as >2 years. It adhered to the Preferred Reporting Items for Systematic Review and Meta-Analysis (S1 Appendix) [35] and the guidelines outlined in the Cochrane Handbook for Systematic Reviews of Interventions [36].

### Eligibility criteria

Inclusion and exclusion criteria are described in Table 1.

### Information sources

An extensive and comprehensive search was conducted from inception to 31st May 2024 in the following key databases:

- MEDLINE (Ovid), Embase (Ovid), Scopus, Web of Science, and CINAHL

- Hand searches of key journals including the *Journal of Arthroplasty*, *the Journal of Bone & Joint Surgery—American Volume*

- Manually screening reference lists of included articles

- Grey literature: Dissertations and Theses of ProQuest (https:// https://www.proquest.com/) and conference proceedings/abstracts in Web of Science and Scopus

### Search strategy

The lead author (SV) in discussion with coauthors and a qualified librarian, developed the search strategy in MEDLINE (Ovid) using the medical subject headings and free text for population, intervention, study design, and outcome measures [31]. The search strategy was then adapted for use in other databases (S2 Appendix). The details defining the keywords were outlined in the protocol [31].

**Table 1. Criteria for inclusion and exclusion of studies.**

| Inclusion criteria | |
|---|---|
| Population | Participants who underwent total hip arthroplasty (THA) due to hip osteoarthritis (OA) |
| Intervention | THA defined as replacing the femoral head and neck, acetabular cartilage and labrum, and subchondral bone by prosthesis [37] with no limitations on the site, type, the approach of THA, and implant being used |
| Comparator | None |
| Outcome measures | Physical measures of spinopelvic alignment: any parameters measuring from spine to the pelvis that contribute to maintaining an energy-efficient posture in normal or pathological status [20,38] (e.g., lumbar lordosis). To accurately characterize spinopelvic alignment, parameters defined by one point on the spine and one on the pelvis were included. This approach ensures that the selected measures specifically reflect the interaction between spinal and pelvic regions, rather than isolated regional parameters.<br>Physical measures of physical functioning:<br>• Impairment-based: any parameter that evaluates abnormal structure or dysfunction in the specific body part or system [39] (e.g., muscle strength).<br>• Performance-based: any parameter that assesses performance on a specific task in a standardized manner [39,40] (e.g., Timed Up and Go test).<br>• Activity level in natural environment: any parameter that evaluates activity in a natural environment [40] (e.g., daily steps)<br>Patient-reported outcome measures (PROMs): any outcome measure that is reported by patients [41]. These include PROMs for low back pain – pain and discomfort between the regions below the costal margin and above the inferior gluteal folds, characterized with or without leg pain [42]. |
| Design | Observational studies including retrospective and prospective longitudinal cohorts and cross-sectional studies which measured the associations following THA with no restriction on the language and the type of association (e.g., concurrent, predictive, responsiveness, etc.). |
| Exclusion criteria | |
| Population | Surgery due to secondary hip OA, developmental pathology, inflammatory disorders, infection, avascular necrosis, trauma, tumor in the hip joint, surface replacement arthroplasty, and revision THA |
| Outcome measures | Clinician-based questionnaires |

## Study records

**Data management.** Covidence, a web-based software to manage and streamline systematic reviews, was used to store citations [43]. It detected and removed the duplicates. After screening the titles and abstracts, full texts of eligible studies were uploaded for full-text screening stage.

**Selection process.** Two authors (SV and SS) independently screened titles and abstracts against the eligibility criteria. Full-texts were obtained if studies met the eligibility criteria or were ambiguous after initial screening. If the full-text was not available, an email was sent to authors, followed by a reminder after two weeks. If there was still no response, the abstract was reported as not retrieved. To confirm the eligibility criteria, authors were contacted to verify the THA indication or to request separate THA results when studies also included total knee arthroplasty or hip resurfacing arthroplasty. A reminder was sent after two weeks; if there was still no response, the study was excluded. Non-English studies were translated in three stages: requesting English versions from corresponding authors, using ChatGPT for translation if the English version was unavailable and asking the authors for proofreading the translated version. If the authors did not respond, native speakers were sought through the research team to proofread the translated version. We selected ChatGPT over Google Translate for its better preservation of stylistic elements, contextual nuances, and strong performance in medical translations [44]. However, as noted in previous studies, this approach may have some limitations [44,45]. To ensure accuracy, all translations in our study were reviewed by native speakers of the respective languages. Two authors (SV and SS) then performed full-text screening. A third reviewer (AR) was available to resolve discrepancies if agreement was not achieved. Agreement was evaluated using Cohen's kappa [46].

**Data collection process.** Three authors (SV, RR, LF) independently extracted data on study characteristics, participants, and outcome measures, with details provided in the protocol [31], using standardized forms. We had two

reviewers act as the second reviewer. One reviewer (SV) conducted the process for half of the studies in collaboration with RR and the other half with LF. They initially piloted the process with five articles and discussed modifications. A third reviewer (AR) mediated if agreement was not achieved. If data were unclear or missing, corresponding authors were contacted, with two follow-up reminders at two-week intervals. If there was no response, the study was reported in the results.

## Risk of bias assessment

Three reviewers (SV, RR, LF) assessed the risk of bias (RoB) using the National Institutes of Health (NIH) quality assessment tool for observational cohort and cross-sectional studies [47]. The reviewers first evaluated RoB of five studies as a pilot and discussed modifications. Discrepancies were discussed to reach a consensus, with a third reviewer (AR) available to mediate if required. Agreement was evaluated using Cohen's kappa [46]. Based on the scoping search to develop the review protocol, the NIH tool was selected over the Newcastle-Ottawa Scale for its more comprehensive bias assessment across nine domains (e.g., potential confounding variables and power analysis) [48]. The best approach to using the NIH tool involves considering how questions may contribute to potential bias in a study [48]. Based on the study's objectives and eligibility criteria, five questions were removed, following common adaptations of the NIH tool [49,50]. The question regarding exposure measured prior to the outcome was removed because it was consistently answered "yes" in all studies according to our eligibility criteria. Questions on exposure levels, assessing the exposure more than once, timeframe sufficiency, and assessor blinding were not applicable. With nine questions remaining, the NIH was rated as good with 8–9 yes answers (yes answers >80%), fair with 6–7 yes answers (yes answers between 60–80%), and poor with less than 5 yes answers (yes answers <60%) [51].

## Data synthesis

Clinical and statistical heterogeneity precluded a meta-analysis, and a narrative synthesis was conducted by adapting Cochrane [52], aligned with the a priori protocol [31]. The synthesis considered study design, RoB, population categories (with and without LBP), outcome measures, statistical parameters, and follow-up timepoints [52]. Postoperative timelines were categorized as short-term (≤3 months), medium-term (>3 months to ≤2 years), and long-term (>2 years) [33,34,53,54]. Associations were synthesized and reported qualitatively when ≥2 studies had similar design, population, outcome measures, type of association, and follow-up timepoints. When narrative synthesis of the findings was not possible, significant results from individual studies were reported. These were defined as associations between measures with p-values <0.05 and correlation coefficients greater than 0.6, which indicate a moderate to perfect association [55]. In regression analysis, the significant finding has been defined as the associations with p-values <0.05 [56,57].

## Meta-bias (es)

To evaluate reporting bias, protocols of included studies were sought, and if available, consistency between the protocol and published results was assessed.

## Confidence in cumulative evidence

Grading of Recommendations Assessment, Development, and Evaluation (GRADE) was used to rate the overall quality of evidence, considering RoB, inconsistency, imprecision, indirectness, and publication bias. The framework was adapted for prognostic studies [58,59], with the assessment of publication bias guided by methods appropriate for observational study designs [60]. GRADE initially assigned low-quality to each category due to inclusion of observational studies [61]. Evidence was downgraded for high/moderate-RoB, inconsistent findings, imprecision, indirectness, and publication bias, but upgraded for large effect sizes or exposure-response relationships. The overall evidence quality was rated as high,

moderate, low, or very low [58]. Three reviewers (SV, RR, LF) rated GRADE, discussing disagreements to reach a consensus. Any discrepancies were mediated by a third reviewer (AR).

## Patient and public involvement

The review process was discussed with the Spinal Pain Research Patient Partner Advisory Group at the School of Physical Therapy, Western University, to incorporate their feedback and enhance its quality.

## Results

### Study selection

Searches in key databases retrieved 23,808 records. In addition, grey literature searches and searching key journals retrieved 2,302 records. After removing duplicates, 12493 titles and abstracts were screened, with 352 meeting criteria for full-text screening. Detailed information on eligibility confirmation, including full-text article retrieval, verification of THA indication, and requests for separate THA results, has been provided in S3 Appendix. Twenty-nine full-text articles could not be retrieved, and seven were available as conference abstracts or abstracts of the included main articles, leaving a total of 316 full-texts for full-text screening. Sixteen non-English studies were screened at the full-text stage; one author provided a translated version [62] and 15 studies were translated. Fifty-one studies (53 articles) met eligibility criteria, with two studies each consisting of two articles [9,63]. Complete consensus reached at both titles and abstracts screening and full-text screening stages, and no mediation required from the third reviewer. Study selection processes and reasons for exclusion (S4 Appendix) have been provided in PRISMA flow diagram (Fig 1).

### Study characteristics

The characteristics of the included studies are presented in S5 Appendix.

### Methods

The 51 studies were conducted in 15 countries, with USA (n = 13) and Japan (n = 11) the most common. Studies were published between 2000 and 2024, 49 studies were in English, 1 in Japanese [64] and 1 in Chinese [65]. There were 17 cross-sectional [22,66–81], 27 cohort longitudinal [9,10,12,19,24,25,63,64,82–100], and 7 studies with both cross-sectional and cohort longitudinal designs [21,65,101–105]. Follow-up periods ranged from one day to 10 years following THA, with most studies focusing on the medium-term (n = 25 studies) [10,12,19,24,25,65,68,71–73,77,79,80,82–84,86,88,90,93,95,97,99–101] and short-term (n = 17 studies) [9,21,22,63,64,67,85,87,89,91,92,96,98,102–105] follow-up. Only 7 studies [66,70,74,76,78,81,94] had long-term follow-up. Two studies' follow-up period could not be determined due to including two timepoints: short-term and medium-term follow-up in one study [69] and medium-term and long-term follow-up in the other [75]. There was missing information in one of the included studies [74], and was unaddressed after contacting the author.

### Participants

A total of n = 4106 participants was included across studies, with sample sizes ranging from 8 to 1383 and age ranging from 52.46 to 76.20. THA was unilateral in most studies (n = 34), and three studies included both unilateral and bilateral procedures. A lateral THA approach was used most frequently (n = 10), followed by an anterolateral approach (n = 9) and a posterolateral approach (n = 8). Only three studies included participants with LBP [19,82,94].

### Physical measures of spinopelvic alignment

Spinopelvic alignment measures were investigated in n = 6 studies, including lumbar lordosis, slip angle, spino–sacral angle, sagittal vertical axis, and T1 pelvic angle (T1PA).

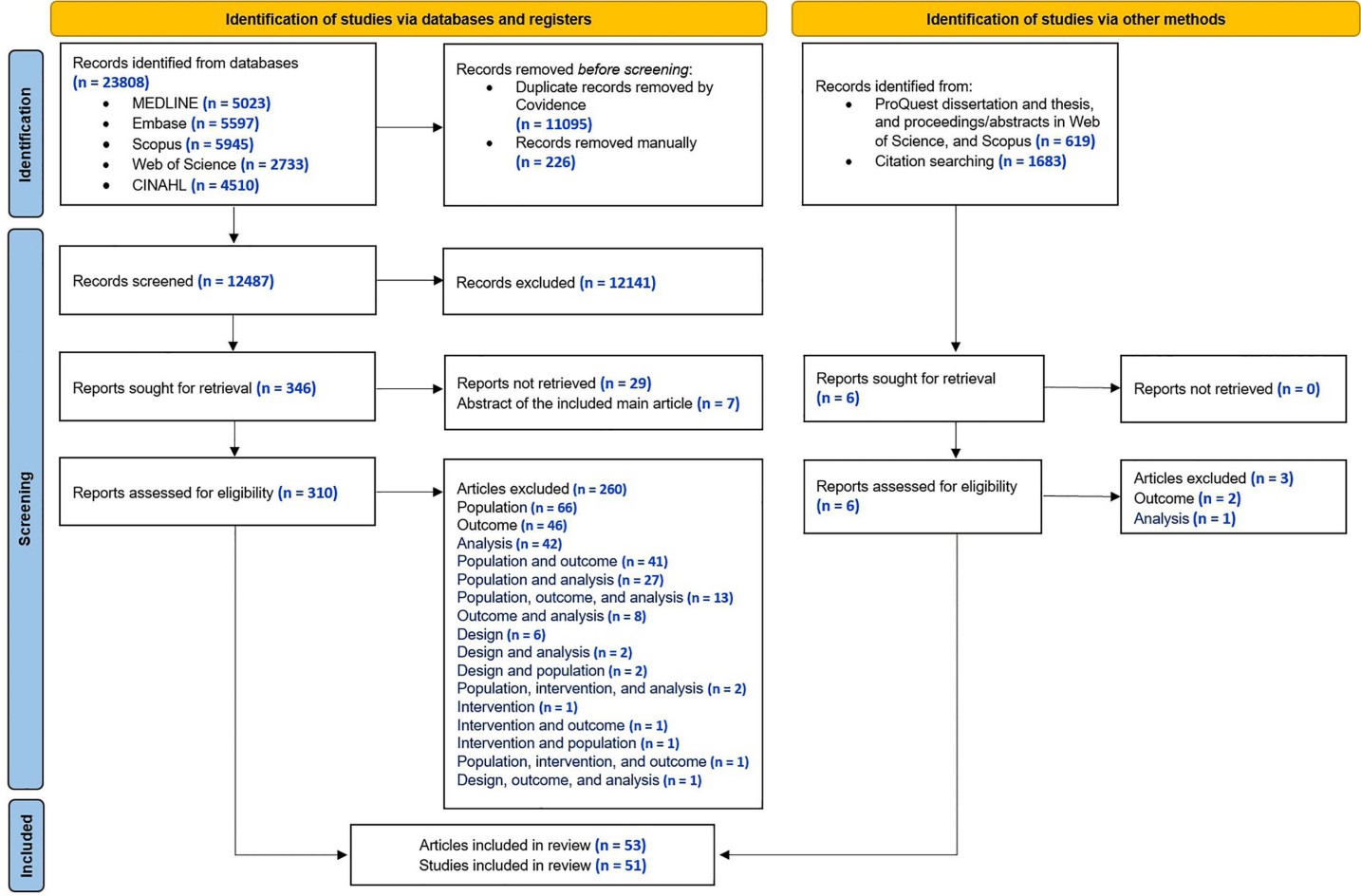

**Fig 1. PRISMA flow diagram.**

## Physical measures of physical functioning

The most common physical measures investigated were impairment-based in n = 27 studies (gait parameters, walking speed, hip kinetics and kinematics parameters, trunk movement variability, weight-bearing chair-rising test, ambulation speed test, balance parameters, hip range of motion (RoM), hip flexion contracture, hip and knee muscle strength, hand-grip strength, and quadriceps performance). Performance-based outcome measures were investigated in n = 15 studies (6 Minute Walk Test (6MWT), Timed-Up-and-Go (TUG), Chair Stand Test (CST), Stair Climb Test (SCT), 40 meter fast-paced-walk test (40 m FPWT), and 10 meter walk test (10 MWT)). Activity level in natural environment measures were investigated in n = 14 studies (habitual speed, daily steps, number of standings, number of sit-to-stand transitions, upright time, physical activity, time sitting, time standing, time walking, time cycling, total time active, cadence, number of intensity peaks < 2.0g, and number of intensity peaks > 2.0g).

## PROMs

The most common PROM constructs investigated were function (n = 18 studies), hip pain (n = 12 studies), pain (n = 7 studies), QoL (n = 7 studies), and physical activity (n = 5 studies). These were followed by fatigue (n = 3 studies), pain

and function (n = 3 studies), clinical outcome (n = 2 studies), fear of falling (n = 2 studies), satisfaction (n = 2 studies), joint awareness (n = 2 studies), and symptoms and function (n = 2 studies). Thirteen PROM constructs were investigated in single studies including disability, knee pain, mental health, health status, hip health, clinical indices, positive feeling, depression, expectation, hip-related health, pain, and function in activities of daily living, functional recovery, pain catastrophizing, and general health: physical functioning.

### LBP

LBP was investigated in only three studies [19,82,94].

### Risk of bias

Eight studies were assessed as low-RoB, 24 as moderate-RoB and 19 as high-RoB (Table 2). Complete agreement in the assessment of RoB was achieved between the 3 reviewers, with and no mediation required from the third reviewer. The question on sample size justification and accounting for the impact of potential confounding variables was rated as high-RoB in most studies. Nearly half of the studies incompletely reported the exposure, including details on the THA approach and type. Some also lacked clarity in defining the study population and participation. Loss to follow-up exceeded 20% in n = 8 studies [9,79,89,93,97,98,100,105] and was unclear [65,74,94] or not reported [10,91] in n = 5 studies. Research questions were clearly defined in all studies, and eligibility criteria were prespecified and applied to all participants. Outcomes were well-defined and implemented in almost all the studies. However, many studies selectively reported results, omitting correlation coefficients, regression coefficients, p-values, or confidence intervals.

### Synthesis across studies

Due to high heterogeneity in measures and follow-up timepoints, a narrative synthesis was conducted. Studies that measured the same type of association between similar outcome measures in similar populations (with or without LBP) and same follow-up timepoints were synthesized. Owing to substantial heterogeneity among studies on the association between spinopelvic alignment with PROMs such as variations in physical measures, PROMs, or follow-up timepoints, it was not possible to conduct a cross-sectional or longitudinal synthesis of the findings. Studies on LBP also could not be synthesized due to heterogeneity. Detailed results of individual studies are presented in S5 Appendix. Using GRADE, the evidence was downgraded in all syntheses for study limitations, imprecision, and publication bias, and for inconsistency in some cases, but not for indirectness (Tables 3 and 4).

### Associations between measures in cross-sectional studies

There was no cross-sectional synthesis on spinopelvic alignment. For physical measures of physical functioning, synthesis was possible for studies on impairment-based outcome measures and activity level in natural environment, but not for performance-based outcome measures.

### Impairment-based outcome measures and PROMs

**Walking speed with patient-reported function in short-term follow-up.** Two studies [68,72] investigated the association between walking speed with patient-reported function in short-term follow-up (n = 2 high-RoB). One study [68] found no association between walking speed with Western Ontario and McMaster Universities Arthritis Index (WOMAC) function at 3 weeks postoperative (postop) and 1 study [72] found no association between walking speed with WOMAC function at 3 months. Very low-level evidence supports no association between walking speed with patient-reported function in short-term follow-up.

**Table 2. Risk of bias of the included studies.**

**Risk of Bias Assessment**

| | Was the research question or objective in this paper clearly stated? | Was the study population clearly specified and defined? | Was the participation rate of eligible persons at least 50%? | Were all the subjects selected or recruited from the same or similar populations? | Was a sample size justification, power description, or variance and effect estimates provided? | Were the exposure measures (independent variables) clearly defined, valid, reliable, and implemented consistently across all study participants? | Were the outcome measures (dependent variables) clearly defined, valid, reliable, and implemented consistently across all study participants? | Was loss to follow-up after baseline 20% or less? | Were key potential confounding variables measured and adjusted statistically for their impact on the relationship? | Summary Quality |
|---|---|---|---|---|---|---|---|---|---|---|
| 1. Luna et al., (2017) 2. Luna et al., (2019) | Yes | Yes | Yes | Yes | No | CD | Yes | No | NR | Poor (5) |
| 1. Heiberg et al., (2013) 2. Heiberg (2013) | Yes | Yes | Yes | Yes | NR | Yes | Yes | Yes | Yes | Good (8) |
| Abujaber (2014) | Yes | Yes | CD | Yes | NR | Yes | Yes | Yes | Yes | Fair (7) |
| Biggs et al., (2022) | Yes | Yes | Yes | Yes | Yes | Yes | Yes | Yes | Yes | Good (9) |
| Boardman et al., (2000) | Yes | Yes | Yes | Yes | NR | CD | Yes | Yes | NR | Fair (6) |
| Bolink et al., (2016) | Yes | Yes | CD | Yes | NR | NR | Yes | Yes | NR | Poor (5) |
| Cao et al., (2022) | Yes | Yes | Yes | Yes | Yes | Yes | Yes | Yes | Yes | Good (9) |
| Casartelli et al., (2015) | Yes | CD | CD | Yes | Yes | CD | Yes | Yes | NR | Poor (5) |
| Cinnamon et al., (2019) | Yes | CD | CD | Yes | Yes | No | Yes | Yes | Yes | Fair (6) |
| Davis et al., (2007) | Yes | Yes | CD | Yes | NR | Yes | Yes | Yes | Yes | Fair (7) |
| Dayton et al., (2016) | Yes | Yes | CD | Yes | NR | Yes | Yes | NR | NR | Poor (5) |
| Eyyazov et al., (2016) | Yes | Yes | CD | Yes | NR | Yes | Yes | Yes | No | Fair (6) |
| Fallahzadeh et al., (2022) | Yes | CD | CD | Yes | NR | NR | Yes | No | NR | Poor (3) |
| Foucher et al., (2010) | Yes | CD | Yes | Yes | NR | CD | Yes | Yes | NR | Poor (5) |
| Foucher et al., (2018) | Yes | CD | Yes | Yes | No | No | Yes | Yes | No | Poor (5) |
| Fujita et al., (2013) | Yes | Yes | Yes | Yes | No | CD | CD | No | Yes | Fair (6) |
| Fujita et al., (2022) | Yes | Yes | Yes | Yes | NR | NR | CD | CD | Yes | Poor (5) |
| Goeb et al., (2021) | Yes | CD | Yes | Yes | NR | Yes | CD | Yes | No | Poor (5) |
| Harada et al., (2024) | Yes | Yes | Yes | Yes | Yes | Yes | Yes | No | Yes | Good (8) |
| Holm et al., (2013) | Yes | Yes | Yes | Yes | NR | Yes | Yes | Yes | No | Fair (7) |
| Holstege et al., (2011) | Yes | Yes | Yes | Yes | Yes | Yes | Yes | No | Yes | Good (8) |
| Huang and Foucher (2019) | Yes | CD | CD | Yes | No | No | Yes | Yes | No | Poor (4) |
| Jelsma et al., (2021) | Yes | Yes | Yes | Yes | NR | Yes | Yes | Yes | NR | Fair (7) |
| Kamimura et al., (2014) | Yes | Yes | Yes | Yes | NR | Yes | Yes | Yes | Yes | Good (8) |
| Kaufmann et al., (2022) | Yes | CD | Yes | Yes | NR | Yes | Yes | Yes | No | Fair (7) |
| Kirschner et al., (2023) | Yes | NR | Yes | Yes | Yes | CD | Yes | Yes | No | Fair (6) |
| Kobayashi et al., (2023) | Yes | Yes | Yes | Yes | NR | Yes | Yes | Yes | No | Fair (7) |
| Lin et al., (2022) | Yes | CD | CD | Yes | NR | CD | Yes | Yes | NR | Poor (4) |
| Lindemann et al., (2006) | Yes | CD | CD | Yes | NR | Yes | Yes | Yes | NR | Poor (5) |
| Lyman et al., (2020) | Yes | Yes | Yes | Yes | NR | CD | Yes | No | NR | Poor (5) |
| Mahmood et al., (2016) | Yes | Yes | Yes | Yes | Yes | Yes | Yes | Yes | Yes | Good (9) |
| Mark-Christensen, Kehlet (2019) | Yes | Yes | CD | Yes | No | CD | Yes | Yes | NR | Poor (5) |

*(Continued)*

Table 2. (Continued)

**Risk of Bias Assessment**

| | Was the research question or objective in this paper clearly stated? | Was the study population clearly specified and defined? | Was the participation rate of eligible persons at least 50%? | Were all the subjects selected or recruited from the same or similar populations? | Was a sample size justification, power description, or variance and effect estimates provided? | Were the exposure measures (independent variables) clearly defined, valid, reliable, and implemented consistently across all study participants? | Were the outcome measures (dependent variables) clearly defined, valid, reliable, and implemented consistently across all study participants? | Was loss to follow–up after baseline 20% or less? | Were key potential confounding variables measured and adjusted statistically for their impact on the relationship? | Summary Quality |
|---|---|---|---|---|---|---|---|---|---|---|
| McMeeken JM, Galea MP (2007) | Yes | CD | CD | Yes | NR | NR | Yes | Yes | NR | Poor (4) |
| Meessen et al., (2020) | Yes | Yes | CD | Yes | NR | CD | Yes | No | Yes | Poor (5) |
| Melchiorri et al., (2015) | Yes | CD | Yes | Yes | NR | Yes | Yes | Yes | NR | Fair (6) |
| Moellenbeck et al., (2020) | Yes | Yes | CD | Yes | NR | NR | Yes | Yes | Yes | Fair (6) |
| Negrini et all., (2020) | Yes | CD | CD | Yes | Yes | Yes | Yes | NR | Yes | Fair (6) |
| Ochi et al., (2017) | Yes | Yes | Yes | Yes | NR | Yes | Yes | Yes | Yes | Good (8) |
| Okamoto et al., (2024) | Yes | Yes | Yes | Yes | NR | Yes | Yes | CD | Yes | Fair (7) |
| Prüfer et al., (2024) | Yes | Yes | Yes | Yes | Yes | NR | Yes | Yes | No | Fair (7) |
| Qiu et al., (2014) | Yes | Yes | CD | Yes | NR | Yes | Yes | CD | NR | Poor (5) |
| Segev–Jacubovski (2023) | Yes | Yes | Yes | Yes | NR | CD | Yes | Yes | Yes | Fair (7) |
| Sliwinski, Sisto (2006) | Yes | CD | CD | Yes | NR | CD | Yes | Yes | NR | Poor (4) |
| Tang et al., (2021) | Yes | CD | Yes | Yes | NR | CD | Yes | No | NR | Poor (4) |
| Tolk et al., (2019) | Yes | Yes | Yes | Yes | Yes | CD | Yes | Yes | NR | Fair (7) |
| Tsukagoshi et al., (2015) | Yes | CD | Yes | Yes | NR | Yes | Yes | Yes | Yes | Fair (7) |
| Tugay et al., (2004) | Yes | Yes | Yes | Yes | NR | Yes | Yes | Yes | No | Fair (7) |
| Vergari et al., (2024) | Yes | Yes | CD | Yes | Yes | NR | Yes | Yes | Yes | Fair (7) |
| Wada et al., (2019) | Yes | CD | Yes | Yes | Yes | Yes | Yes | Yes | No | Fair (7) |
| Wagenmakers et al., (2008) | Yes | Yes | Yes | Yes | NR | NR | Yes | Yes | NR | Fair (6) |
| Yamaguchi et al., (2019) | Yes | Yes | CD | Yes | NR | Yes | Yes | Yes | NR | Fair (6) |

CD: Cannot determined, NR: Not reported.

**Table 3. Overall quality of evidence for cross-sectional studies.**

| Association between measures | Number of participants | Number of studies | Number of associations | Body of evidence | Factors that may reduce the quality | | | | | Factors that may increase the quality | | Overall quality |
|---|---|---|---|---|---|---|---|---|---|---|---|---|
| | | | | | Study limitations | Inconsistency | Indirectness | Imprecision | Publication bias | Moderate/ large effect size | Exposure- response | |
| Walking speed with patient-reported function in short-term follow-up | 62 | 2 | 2 | Low quality | ✗ | ✓ | ✓ | ✗ | ✗ | ✗ | ✗ | Very low quality+ |
| Walking speed with patient-reported function in medium-term follow-up | 62 | 2 | 2 | Low quality | ✗ | ✗ | ✓ | ✗ | ✗ | ✗ | ✗ | Very low quality+ |
| Light intensity physical activity with patient-reported physical activity in medium-term follow-up | 89 | 2 | 4 | Low quality | ✗ | ✓ | ✓ | ✗ | ✗ | ✗ | ✗ | Very low quality+ |
| Total physical activity with patient-reported physical activity in medium-term follow-up | 89 | 2 | 4 | Low quality | ✗ | ✗ | ✓ | ✗ | ✗ | ✗ | ✗ | Very low quality+ |

✗ Serious limitation, ✓ No limitation, ++++ High quality, +++ Moderate quality, ++ Low quality, + very low quality

**Table 4. Overall quality of evidence for cohort longitudinal studies.**

| Association between the measures | Number of participants | Number of studies | Number of associations | Body of evidence | Factors that may reduce the quality | | | | | Factors that may increase the quality | | Overall quality |
| --- | --- | --- | --- | --- | --- | --- | --- | --- | --- | --- | --- | --- |
| | | | | | Study limitations | Inconsistency | Indirectness | Imprecision | Publication bias | Moderate/ large effect size | Exposure-response | |
| Postop TUG with preop hip pain in short-term follow-up | 88 | 2 | 2 | Low quality | ✗ | ✗ | ✓ | ✗ | ✗ | ✗ | ✗ | Very low quality+ |
| Change in TUG with change in pain in short-term follow-up | 31 | 2 | 2 | Low quality | ✗ | ✓ | ✓ | ✗ | ✗ | ✗ | ✗ | Very low quality+ |
| Change in TUG with change in patient-reported function in short-term follow-up | 31 | 2 | 2 | Low quality | ✗ | ✓ | ✓ | ✗ | ✗ | ✗ | ✗ | Very low quality+ |
| Change in CST with change in patient-reported function in medium-term follow-up | 136 | 2 | 2 | Low quality | ✗ | ✗ | ✓ | ✗ | ✗ | ✗ | ✗ | Very low quality+ |
| Change in SCT with change in pain in short-term follow-up | 31 | 2 | 2 | Low quality | ✗ | ✓ | ✓ | ✗ | ✗ | ✗ | ✗ | Very low quality+ |
| Change in SCT with change in patient-reported function in short-term follow-up | 71 | 3 | 3 | Low quality | ✗ | ✓ | ✓ | ✗ | ✗ | ✗ | ✗ | Very low quality+ |
| Change in SCT with change in patient-reported function in medium-term follow-up | 100 | 2 | 2 | Low quality | ✗ | ✓ | ✓ | ✗ | ✗ | ✗ | ✗ | Very low quality+ |

✗ Serious limitation, ✓ No limitation, ++++High quality, +++ Moderate quality, ++ Low quality, +very low quality, Postop: Postoperative, Preop: Preoperative, TUG: Timed Up and Go, CST: Chair Stand Test, SCT: Stair Climb Test

**Walking speed with patient-reported function in medium-term follow-up.** Two studies [68,72] investigated the association between walking speed with patient-reported function in medium-term follow-up (n = 2 high-RoB). One study [68] found no association between walking speed with WOMAC function at 12 months postop, and 1 study [72] found an association between faster walking speed with better WOMAC function at 12 months. Very low-level evidence supports inconsistent findings on the association between walking speed with patient-reported function in medium-term follow-up.

### Activity level in natural environment and PROMs

**Light intensity physical activity with patient-reported physical activity in medium-term follow-up.** Two studies [71,73] investigated the association between light intensity physical activity with patient-reported physical activity in medium-term follow-up (n = 1 high-RoB, 1 moderate-RoB). One study [71] found no association between light intensity physical activity with Physical Activity Scale for the Elderly (PASE) at 4, 7, and 10 months postop, and 1 study [73] found no association between light intensity physical activity with Short Questionnaire to Assess Health-enhancing physical activity (SQUASH) at 1 year. Very low-level evidence supports no association between light intensity physical activity with patient-reported physical activity in medium-term follow-up.

**Total physical activity with patient-reported physical activity in medium-term follow-up.** Two studies [71,73] investigated the association between total physical activity with patient reported physical activity in medium-term follow-up (n = 1 high-RoB, 1 moderate-RoB). One study [71] found no association between total physical activity with PASE at 4, 7, and 10 months postop, and 1 study [73] found an association between better total physical activity with better SQUASH at 1 year. Very low-level evidence supports inconsistent findings on the association between total physical activity with patient-reported physical activity in medium-term follow-up.

**Association between the measures in cohort longitudinal studies.** Due to heterogeneity, studies on spinopelvic alignment could not be synthesized. For physical measures of physical functioning, synthesis was limited to performance-based outcome measures, excluding impairment-based outcome measures and activity level in natural environment.

### Performance-based outcome measures and PROMs

**Postop TUG with preoperative (preop) hip pain in short-term follow-up.** Two studies [84,91] investigated the association between postop TUG with preop hip pain in short-term follow-up (n = 1 low-RoB, 1 moderate-RoB). One study [91] found an association between better performance in TUG at 12 days postop with lower preop hip pain VAS, and 1 study [84] found no association between TUG at 3 weeks postop with preop hip pain VAS. Very low-level evidence supports inconsistent findings on the association between postop TUG with preop hip pain in short-term follow-up.

**Change in TUG with change in pain in short-term follow-up.** Two studies [10,96] investigated the association between change in TUG with change in pain in short-term follow-up (n = 1 moderate-RoB, 1 high-RoB). One study [10] found no association between change in TUG with change in Hip Disability and Osteoarthritis Outcome Score (HOOS) pain from preop to 1 month postop, and 1 study [96] found no association between change in TUG with change in WOMAC pain from 2 weeks preop to 10 weeks. Very low-level evidence supports no association between change in TUG with change in pain in short-term follow-up.

**Change in TUG with change in patient-reported function in short-term follow-up.** Two studies [10,96] investigated the association between change in TUG with change in patient-reported function in short-term follow-up (n = 1 high-RoB, 1 moderate-RoB). One study [96] found no association between change in TUG with change in WOMAC function from 2 weeks preop to 10 weeks postop, and 1 study [10] found no association between change in TUG with change in HOOS ADL from 2 weeks preop to 1 month. Very low-level evidence supports no association between change in TUG with change in patient-reported function in short-term follow-up.

**Change in CST with change in patient-reported function in medium-term follow-up.** Two studies [24,25] investigated the association between change in CST with change in patient-reported function in medium-term follow-up (n = 1 high-RoB, 1 moderate-RoB). One study [25] found an association between greater change in CST with greater change in HOOS physical function from preop to 4 months postop, and 1 study [24] found no association between change in CST with change in HOOS physical function from preop to 1 year. Very low-level evidence supports inconsistent findings on the association between CST with change in patient-reported function in medium-term follow-up.

**Change in SCT with change in pain in short-term follow-up.** Two studies [10,96] investigated the association between change in SCT with change in pain in short-term follow-up (n = 1 high-RoB, 1 moderate-RoB). One study [96] found no association between change in SCT with change in WOMAC pain from 2 week preop to 10 weeks postop, and 1 study [10] found no association between change in SCT with change in HOOS pain from preop to 1 month. Very low-level evidence supports no association between change in SCT with change in pain in short-term follow-up.

**Change in SCT with change in patient-reported function in short-term follow-up.** Three studies [9,10,96] investigated the association between change in SCT with change in patient-reported function in short-term follow-up (n = 2 high-RoB, 1 moderate-RoB). One study [96] found no association between change in SCT with change in WOMAC function from 2 weeks preop to 10 weeks postop, 1 study [10] found no association between change in SCT with change in HOOS ADL from 2 weeks preop to 1 month, and 1 study [9] found no association between change in SCT with change in HOOS ADL from 14 to 3 days preop to 14 days. Very low-level evidence supports no association between change in SCT with change in patient-reported function in short-term follow-up.

**Change in SCT with change in patient-reported function in medium-term follow-up.** Two studies [10,24] investigated the association between change in SCT with change in patient-reported function in medium-term follow-up (n = 1 high-RoB, 1 moderate-RoB). One study [10] found no association between change in SCT with change in HOOS ADL from 1 month to 6 months postop, and 1 study [24] found no association between change in SCT with change in HOOS physical function from preop to 1 year. Very low-level evidence supports no association between change in SCT with change in patient-reported function in medium-term follow-up.

### Significant findings of the individual studies on the association between the physical measures with PROMs

In low-RoB studies, significant associations were found between spinopelvic alignment (sagittal vertical axis and lumbar lordosis) with function [12] and between impairment-based physical measures with PROMs constructs (gait function with change in function, change in gait function with change in pain, function, and symptoms, and knee extensor strength with functional recovery) [83,98]. In moderate-RoB studies, there were significant associations between spinopelvic alignment with PROMs constructs (sagittal vertical axis with knee pain and T1PA with QoL) [85,95] and between impairment-based outcome measures with PROMs constructs (change in dynamic hip RoM during the stance phase of gait with change in symptoms, hip abductor muscle strength of the operated side at −5 degree, 10 degree, 15 degree, and 20 degree with QoL, hip abductor strength of the operated side with function, peak hip extension angle with hip pain, and hip motion and RoM with function) [66,75,78,81,99]. There were also significant association in moderate-RoB studies between performance-based outcome measures with PROMs constructs (TUG with QoL, change in TUG, SCT, and 6MWT with change in hip pain) [78,104] and between activity levels in natural environment with PROMs constructs (Moderate intensity activity with QoL, sedentary break with disability, change in daily steps with change in fear of falling, mean physical activity with patient-reported total physical activity, time walking with joint awareness, time walking, total time active, and numbers of steps with physical function, time walking, total time active, and numbers of steps with general health: physical functioning) [64,73,76,79,90].

### Significant findings of the individual studies on the association between the physical measures with LBP

There was significant associations between spinopelvic alignment including T1PA with PROMs of LBP [94].

## Reporting bias

The protocols of 9 studies were identified. While 6 studies followed their registered protocols [25,72,86,87,90,99], 2 studies had deviations without any explanation [91,96]. In 1 study, additional outcomes were measured compared to the protocol with no explanation provided [9].

## Patient and public involvement

After presenting the findings to the patient partners, they rated the clarity as very to extremely clear, the alignment between key findings and conclusions as somewhat to extremely clear, and the potential impact as good to excellent, with one rating it as fair. Patients also highlighted the importance of pelvic alignment in understanding physical function and outcomes following THA. Additionally, some also noted hypermobility as a potential confounding factor to consider when interpreting the findings. While we could not incorporate their feedback directly, due to our focus on spinopelvic rather than pelvic parameters, and the heterogeneity that prevented synthesis to assess the effects of hypermobility, we have included these points as recommendations for future research.

## Discussion

The key finding from this review was the very low-quality evidence showing no association or inconsistent association between physical measures of physical functioning with PROMs, and no evidence on the association between spinopelvic alignment with PROMs. The association between physical measures with LBP was examined in only 3 studies, with different results. The RoB was concerning, with only 8 studies rated as low-RoB. The evidence is at an early stage, with significant heterogeneity in outcomes. Thus, the current understanding of the association between measures is limited.

### Association between physical measures with PROMs

There was no synthesized evidence on the association between the physical measures of spinopelvic alignment with PROMs. Only six studies have evaluated this relationship. However, in the scoping review by Pourahmadi et al. [20] on spinopelvic alignment and LBP following THA, more studies were identified. This discrepancy may reflect differences in how spinopelvic parameters were defined. In this review, spinopelvic alignment was narrowly defined as involving one parameter on the spine and one on the pelvis, which led to the exclusion of studies assessing pelvic parameters alone. The objectives also differed. Pourahmadi et al. [20] aimed to provide an overview of spinopelvic alignment and LBP following THA, whereas this review specifically focused on studies evaluating the association between spinopelvic alignment and LBP following THA. The limited number of studies addressing association further highlights a gap in the existing literature. This evidence gap limits our ability to draw firm conclusions about the influence of spinopelvic alignment on PROMs and underscores the need for standardized measurement approaches and systematic evaluation to enable more robust synthesis and clinical interpretation. Very low-quality evidence supports no association or inconsistent association between physical measures of physical functioning with PROMs e.g., walking speed with patient-reported function in short-term follow-up or TUG change with pain/function change in short-term follow-up; likely due to several factors, including the lack of low-RoB studies, inadequate sample sizes [106], and the presence of confounding variables [107]. Variations in surgical approaches might also influence the THA outcomes [108–110], with anterior approach showing earlier recovery of function and less pain than posterior or lateral approaches at short-term follow-up [109,110] due to less muscle splitting and soft tissue damage [111,112]. The included studies indicate that the lateral approach was used most frequently, followed by the anterolateral and posterolateral approaches, with some studies not specifying the approach used. Differences in follow-up timepoints, with most studies focusing on medium- and short-term follow-up, may further impact results as recovery processes for physical measures and PROMs may vary [7,113], depending on the surgical approach [114–116]. These findings contradict previous reviews that showed an association between preoperative function or pain

with clinical or functional outcomes following THA [26–30]. This difference may be due to their examination of associations without restricting the type of outcomes, whether physical measures or PROMs [26–30], restricting their PROMs to pain and/or function [28], or focusing on the studies with short-term follow-up ≤2 weeks [30]. Additionally, the lack of consistent findings in our review may be influenced by statistical heterogeneity, as we included studies examining any type of association.

## Association between physical measures with LBP

There was no synthesized evidence on the association between the physical measures with LBP, and only 3 studies investigated their relationship, reporting inconsistent findings with methodological limitations. Eyvazov et al. found LBP improvement was not associated with change in spinopelvic alignment (sagittal vertical axis) or impairment-based measures (e.g., postural stability test) [19], likely due to the lack of clarity on sample size calculation, potentially leading to an underpowered study, or because they may not consider the effects of confounding variables [106,107]. Cao et al. found no association between slip angle and spino-sacral angle with LBP, but an association between increased lumbar lordosis with lower LBP severity [82]. Although this study rated as low-RoB, its findings were presented only based on the significance of p-value, which cannot reveal the strength of the association [55]. An association was also noted between preoperative T1PA with postoperative LBP [94]; However, limitations such as not reporting the sample size calculation were evident, which may affect the study's power [106]. These findings aligned with the review by Pourahmadi et al. which highlighted inconsistent results and methodological issues in studies on spinopelvic alignment and LBP following THA [20]. Currently there is limited insight into the relationship between LBP and physical measures within this population.

## Heterogeneity across studies

The main factor for heterogeneity was different outcome measures across the studies. Despite more than a decade since OARSI recommended a set of performance-based measures in individuals with hip OA [11], relatively few studies published after the guideline have used the recommended measures. This aligns with previous reviews noting significant variations in outcome measures for total hip or knee arthroplasty [117–119]. The same lack of standardization applied to the PROMs. Although the systematic review by Gagnier et al. [120] identified PROMs that assess domains of interest to researchers and allowed them to select the instrument with the best properties, the included studies measured a wide range PROMs. Notably, almost half of the studies in our review were published before Gagnier et al.'s review [120]. These findings were consistent with other reviews highlighting substantial variability in PROMs in THA clinical trials [121], or studies on total hip or knee arthroplasty [122,123].

## Limitations and strengths

This systematic review is the first to synthesize evidence on the association between physical measures of spinopelvic alignment and physical functioning with PROMs and the association between the physical measures with LBP. By addressing the significant heterogeneity of the measures and the lack of high-quality evidence, it highlights important gaps in literature, particularly the limited studies on LBP. Since no limitations were placed on the type of association (e.g., concurrent, predictive, responsiveness, etc.), this methodological heterogeneity may contribute to the observed inconsistencies in the findings. The findings are specific to THA due to hip OA and may not be generalizable to other THA indications.

## Recommendations for future research

An adequately powered, low-RoB study to evaluate the association between the physical measures of spinopelvic alignment and physical functioning with PROMs, and the association between the physical measures with LBP is required. Further research is also needed on long-term follow-up, focusing on outcome measures with potential value yet to be

investigated, e.g., muscle activity or biomechanical parameters during functional activities. Different surgical approaches may also influence the association between the measures and warrant further investigation. It would also be important for future reviews to examine the relationship between spinopelvic alignment and pelvic alignment including implant positioning or component orientation, and how these are associated with functional outcomes. Furthermore, assessing the impact of hypermobility on these associations would be valuable.

## Conclusion

Very low-quality evidence supports no association or inconsistent findings on the relationship between physical measures of physical functioning with PROMs. A key finding of this review is that the limited and heterogeneous nature of available studies on the association between spinopelvic alignment and PROMs prevented meaningful synthesis, highlighting a clear gap in existing literature. Three studies investigated the relationship between the measures with LBP, yielding varied results that were not possible to synthesize. Heterogeneity was evident with few low-RoB studies. This highlights the need for further research investigating the association between the measures in adequately powered, low-RoB studies. Additionally, more prospective studies are needed to investigate validated spinopelvic parameters and to better control for lumbar spine pathology, such as LBP.

## Supporting information

**S1 Appendix. PRISMA checklist.**
(DOCX)

**S2 Appendix. Search strategies.**
(DOCX)

**S3 Appendix. Eligibility confirmation.**
(DOCX)

**S4 Appendix. Studies identified in literature search.**
(DOCX)

**S5 Appendix. Study characteristics and results.**
(DOCX)

## Acknowledgments

We thank Maren Goodman (Librarian) for helping us in developing our search strategy.

## Author contributions

**Conceptualization:** Sima Vatandoost, Brent Lanting, K.C. Geoffrey Ng, Katie Kowalski, Alison Rushton.

**Formal analysis:** Sima Vatandoost.

**Investigation:** Sima Vatandoost, Saghar Soltanabadi, Renee-Marie Ragguett, Lucas Fernandino.

**Methodology:** Sima Vatandoost, Saghar Soltanabadi, Renee-Marie Ragguett, Lucas Fernandino, Brent Lanting, K.C. Geoffrey Ng, Katie Kowalski, Alison Rushton.

**Project administration:** Sima Vatandoost.

**Supervision:** Alison Rushton.

**Visualization:** Sima Vatandoost.

**Writing – original draft:** Sima Vatandoost.

**Writing – review & editing:** Sima Vatandoost, Saghar Soltanabadi, Renee-Marie Ragguett, Lucas Fernandino, Brent Lanting, K.C. Geoffrey Ng, Katie Kowalski, Alison Rushton.

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
