## [Decision Letter · Decision Letter 0]

16 May 2025

Dear Dr. Vatandoost,

Thank you for submitting your manuscript to PLOS ONE. After careful consideration, we feel that it has merit but does not fully meet PLOS ONE’s publication criteria as it currently stands. Therefore, we invite you to submit a revised version of the manuscript that addresses the points raised during the review process.

We look forward to receiving your revised manuscript.

Kind regards,

Masaya Anan, Ph.D.

Academic Editor

PLOS ONE

Journal Requirements:

3. As required by our policy on Data Availability, please ensure your manuscript or supplementary information includes the following:

Reviewers' comments:

Reviewer's Responses to Questions

**Comments to the Author**

1. Is the manuscript technically sound, and do the data support the conclusions?

Reviewer #1: Yes

Reviewer #2: Yes

2. Has the statistical analysis been performed appropriately and rigorously?

Reviewer #1: Yes

Reviewer #2: Yes

3. Have the authors made all data underlying the findings in their manuscript fully available?

Reviewer #1: Yes

Reviewer #2: Yes

4. Is the manuscript presented in an intelligible fashion and written in standard English?

Reviewer #1: Yes

Reviewer #2: Yes

Reviewer #1: Thank you for the opportunity to review this systematic review on the association between physical measures—especially spinopelvic alignment—and patient-reported outcomes after total hip arthroplasty (THA). The paper is well-written, thorough, and addresses a very relevant and emerging topic in joint arthroplasty.

That said, I do have a few comments and suggestions that I hope can help strengthen the manuscript further:

The title highlights spinopelvic alignment, but only six of the included studies actually looked at spinopelvic measures. This might be a bit confusing for readers who expect a more focused synthesis on that topic.

It would be helpful if the authors could discuss this in the Introduction and/or Discussion. Was this due to a true lack of studies in the field? Or were there strict inclusion criteria that limited the number of relevant studies? Either way, a short explanation would help set expectations.

Since spinopelvic alignment can influence functional outcomes and has been linked to implant positioning—especially personalized cup alignment—it would be good to expand on that point. For example, did any of the included studies look at how spinopelvic mobility (e.g., stiff vs flexible pelvis) affected outcomes or guided component orientation?

Consider briefly discussing whether any of the reviewed studies addressed these concepts, or mention this as an area for future research.

The conclusions accurately reflect the overall limitations in the literature. However, this paper can also serve as a useful roadmap for future work in this area.

Consider adding a few specific recommendations, such as: The need for standardized outcome measures.

More prospective studies using validated spinopelvic parameters. Better control for variables like lumbar spine pathology or pelvic stiffness.

Reviewer #2: General comments:

The authors present a systematic review article on the relationship between patient-reported outcomes (PROMs) and spinopelvic alignment or physical functioning on total hip arthroplasty patients. Secondary objective was to study the relationship of physical measures with lower back pain. 51 studies in total were included to the review. The main finding was that there was no clear association between specific physical function measurements and PROMs but the quality of studies was low. Similarly, there were only 3 studies that reported lower back pain and no clear outcomes could be reported.

The study methods on gathering data are structured and scientifically sound. The study design was well constructed. The criticism and suggestions for improvements are mainly focused on the reporting of the data. While there is no word limit on the articles, the main text and the tables contain over 21000 words, which is way higher number than in most of the articles. Therefore, I recommend shortening the text and the tables, and possibly including rest of the data as a supplementary data. I also think that it is unnecessary to have multiple objectives on the review article (association of PROMs to three different outcomes).

Specific comments:

Objectives

- Lines 61-2: “For THA, physical measures of spinopelvic alignment and physical function are important”. This line should be elsewhere, in the next paragraph where the authors describe the spinopelvic alignment.

- Line 88-89: Is it necessary to report lower back pain as a separate secondary objective or could it be part of the other PROMs?

Materials and methods

- Table 1. The authors mention three outcome measures for inclusion to the review: 1) physical measures of spinopelvic alignment 2) physical measures of physical functioning and 3) PROMs. Later, they find that the studies for spinopelvic alignments and relationship to PROMs are scarce and the main focus stays on the physical functioning measurements. While this is a result as itself, would it be clearer if the article focused only to the physical functioning measurements? Alternatively, the title should be changed so that it emphasizes the physical functioning.

- Line 144: ChatGPT was used in the translation of the foreign language text. Although the translation probably works quite well, I’ m not sure if it has been validated as a proper translation of scientific texts.

- Lines 154 and 163: “a third reviewer”. You mean fourth? There was a mention of three authors who extracted data and that there was a separate mediator.

- Line 176: reference to [49] Pais et al (2020). How is the text related to the reference?

- Tables 3 and 4 are clear.

Results

- Lines 221-236. I think there is way too much text. I find it difficult to follow the narrative. A simple flow chart is enough.

- Table 2. This table has a summary of every article included to the review including name, publication year, study objectives, number of patients and sex, health conditions physical outcome measures, PROMs, follow-up time and summary of results. While its impressive, I find it very difficult to read, mainly due the length of up to 40 pages and the extensive use of abbreviations (footnotes are 15 lines long). Overall, its impossible to form a coherent summary of the results. I think it would be more beneficial if this table was transferred to supplementary data or to the end of the article and this table greatly shortened to include only key results of the each stydy.

- Lines 209-310: Again, I don’t think it’ s necessary to report lower back pain separately.

- Lines 403-418: The results here are confusing related to the methods of the article. In other studies which have been included to the review, the authors study the correlation between PROMs and actual measured physical functioning. If I have understood correctly, these measurements have been taken at the same time. However, the studies ion this paragraph ([82, 93, 10, 95]) focused on the preoperative measurements and how they predict postoperative outcomes, which in my opinion is a completely different kind of correlation. The authors could describe more clearly what kind of time-intervals were allowed when studying the correlation of PROMs and functional measurements or remove these studies.

- Lines 498-502: I think this paragraph is not needed.

Discussion

- Lines 515-516: In the methods it is stated that the objective was to study 1) spinopelvic alignment 2) physical functioning measures and 3) lower back pain and their relationship to PROMs. However, the spinopelvic alignment is barely discussed later on, possibly mainly because of the lack of studies. Again, my suggestion is to either remove the spinopelvic alignment altogether from the methods and focus on the physical functioning or add a proper paragraph to the discussion section.

- Lines 529-530: Again, see my comment on lines 403-418.

Overall, I recommend the publication of the article but needs some revisions regarding the focusing of the study design. The results section should also be shortened especially regarding the table 2.

**Do you want your identity to be public for this peer review?** For information about this choice, including consent withdrawal, please see our Privacy Policy

Reviewer #1: No

Reviewer #2: No

---

## [Author Response · Author response to Decision Letter 1]

6 Jul 2025

Reviewers' comments: Reviewer #1: Thank you for the opportunity to review this systematic review on the association between physical measures—especially spinopelvic alignment—and patient-reported outcomes after total hip arthroplasty (THA). The paper is well-written, thorough, and addresses a very relevant and emerging topic in joint arthroplasty.

Response to reviewers: Thank you so much for your feedback on the content and topic of the systematic review. It’s encouraging to know that the time we invested in developing the topic and conducting the study was worthwhile.

Reviewers' comments: The title highlights spinopelvic alignment, but only six of the included studies actually looked at spinopelvic measures. This might be a bit confusing for readers who expect a more focused synthesis on that topic.

It would be helpful if the authors could discuss this in the Introduction and/or Discussion. Was this due to a true lack of studies in the field? Or were there strict inclusion criteria that limited the number of relevant studies? Either way, a short explanation would help set expectations.

Response to reviewers: Thank you for your valuable feedback. Studying the relationship between physical measures of spinopelvic alignment and PROMs was one of the main focuses of this study, developed based on the scoping search, through which we defined the study objectives. Our inclusion criteria required parameters to include one point on the spine and one on the pelvis to focus exclusively on the spinopelvic region. This resulted in six included studies, representing the total available based on our definition.

We have incorporated the following context into the introduction to set clear expectations for the reader upfront, which we believe addresses your concern about potential confusion before reaching the results or discussion sections:

-Line 73-77: “Although interest in spinopelvic alignment and THA outcomes is growing, systematic reviews focusing specifically on physical measures linking spinal and pelvic reference points remain limited. To address this gap, we applied strict inclusion criteria, requiring measures with one point on the spine and one on the pelvis, resulting in a smaller but more targeted pool of studies, further discussed in the review.”

We also added the following explanation in the discussion section:

- Line 514-518: “Only six studies have evaluated this relationship. One possible reason for the limited number of studies in this area may be the stringent inclusion criteria used in this review, which precisely required parameters with one point on the spine and one point on the pelvis, thereby focusing on a specific definition of spinopelvic alignment relevant to biomechanical considerations.”

Reviewers' comments: Since spinopelvic alignment can influence functional outcomes and has been linked to implant positioning—especially personalized cup alignment—it would be good to expand on that point. For example, did any of the included studies look at how spinopelvic mobility (e.g., stiff vs flexible pelvis) affected outcomes or guided component orientation?

Consider briefly discussing whether any of the reviewed studies addressed these concepts, or mention this as an area for future research.

Response to reviewers: Thank you for highlighting the role of spinopelvic alignment in influencing functional outcomes and its link to implant positioning, particularly personalized cup alignment. While these are important factors, the objective of our review was specifically to evaluate the relationship between spinopelvic alignment, not mobility (e.g., stiff vs. flexible pelvis), and patient-reported outcome measures (PROMs). Assessing the relationship between spinopelvic mobility and functional outcomes or implant positioning was outside the current review's scope; therefore, none of the included studies looked at the relationship between spinopelvic mobility with functional outcomes or factors such as cup alignment or component orientation. Nonetheless, we have identified this as an important area for future research and have noted it in the manuscript with the following statement:

- Line 585-587: “It would also be important for future reviews to examine the relationship between spinopelvic alignment and pelvic alignment including implant positioning or component orientation, and how these are associated with functional outcomes.”

Reviewers' comments: The conclusions accurately reflect the overall limitations in the literature. However, this paper can also serve as a useful roadmap for future work in this area.

Consider adding a few specific recommendations, such as: The need for standardized outcome measures.

More prospective studies using validated spinopelvic parameters. Better control for variables like lumbar spine pathology or pelvic stiffness.

Response to reviewers: Thank you for the feedback regarding the conclusion and its reflection on the limitations in literature. We have dedicated a paragraph within the discussion to explain the heterogeneity and the need for standardized outcome measures. As this point is thoroughly covered in the discussion section, we did not repeat it in the recommendations for future research to avoid redundancy. A minor edit was made to this paragraph for clarity.

We have revised the final sentence of the conclusion to emphasize the need for more prospective studies using validated spinopelvic parameters, as follows:

- Line 598 and 599: “Additionally, more prospective studies are needed that investigate validated spinopelvic parameters and better control for lumbar spine pathology, such as LBP.”

Reviewers' comments: Reviewer #2: General comments:

The authors present a systematic review article on the relationship between patient-reported outcomes (PROMs) and spinopelvic alignment or physical functioning on total hip arthroplasty patients. Secondary objective was to study the relationship of physical measures with lower back pain. 51 studies in total were included to the review. The main finding was that there was no clear association between specific physical function measurements and PROMs but the quality of studies was low. Similarly, there were only 3 studies that reported lower back pain and no clear outcomes could be reported. The study methods on gathering data are structured and scientifically sound. The study design was well constructed. The criticism and suggestions for improvements are mainly focused on the reporting of the data. While there is no word limit on the articles, the main text and the tables contain over 21000 words, which is way higher number than in most of the articles. Therefore, I recommend shortening the text and the tables, and possibly including rest of the data as a supplementary data. I also think that it is unnecessary to have multiple objectives on the review article (association of PROMs to three different outcomes).

Response to reviewers: We greatly appreciate your feedback on the study’s methodology and design. We have carefully considered your comments regarding the word count. To address this, we have provided table 2 on study characteristics and results as a supplementary file. Additionally, we have included supplementary material detailing the results of the eligibility confirmation process at the full-text review stage. These changes have significantly reduced the manuscript’s length: prior to revision, the word count from the introduction to the acknowledgements was 22,424 words; following these revisions, the total word count has been reduced to 7,135, including words in tables and in-text references.

This review was structured with two distinct objectives to address two separate and clinically relevant questions. While both involve physical measures, one focuses on their relationship with PROMs and the other on their relationship with low back pain (LBP). Given the inconsistent reporting in the literature and the limited ability of general PROMs to adequately capture the relationship between these physical measures and LBP, we considered a separate objective and analysis necessary to ensure clarity and precision in this context.

Reviewers' comments: Specific comments:

Objectives

- Lines 61-2: “For THA, physical measures of spinopelvic alignment and physical function are important”. This line should be elsewhere, in the next paragraph where the authors describe the spinopelvic alignment.

Response to reviewers: Your specific feedback is much appreciated.

- Lines 61-2: This line has been moved at the beginning of the following paragraph, where the explanation of spinopelvic alignment is provided.

Reviewers' comments: - Line 88-89: Is it necessary to report lower back pain as a separate secondary objective or could it be part of the other PROMs?

Response to reviewers: - Line 88-89: We appreciate your insightful comment. While LBP can be considered a component of PROMs, we chose to report it as a separate secondary objective due to its high prevalence and specific clinical relevance in the THA population. Additionally, our scoping search revealed that the relationship between physical measures of spinopelvic alignment and physical functioning with LBP has been infrequently and inconsistently reported in the literature. Given that general PROMs may not sufficiently capture the relationship between these physical measures and LBP, we believed it warranted focused analysis as a separate outcome to better understand its role in this context.

Reviewers' comments: Materials and methods

- Table 1. The authors mention three outcome measures for inclusion to the review: 1) physical measures of spinopelvic alignment 2) physical measures of physical functioning and 3) PROMs. Later, they find that the studies for spinopelvic alignments and relationship to PROMs are scarce and the main focus stays on the physical functioning measurements. While this is a result as itself, would it be clearer if the article focused only to the physical functioning measurements? Alternatively, the title should be changed so that it emphasizes the physical functioning

Response to reviewers: We sincerely appreciate your insightful feedback. Our aim was to provide a comprehensive overview by including both physical functioning measures and spinopelvic alignment. While the narrative synthesis showed less heterogeneity in studies on physical functioning, this does not reflect a narrowed focus. Instead, the limited number and heterogeneity of spinopelvic studies made firm conclusions difficult, though they represent the full scope of available evidence. This highlights a critical gap and the need for further research in this area. Additionally, the research question was defined based on a preliminary scoping search, and the review was conducted in accordance with a predefined, published, and PROSPERO-registered protocol. As such, altering the study objectives at this stage could introduce bias and compromise the methodological integrity of the review. For further clarification, the following statement has been added to the synthesis paragraph in the results section:

- Line 320-324: “Owing to substantial heterogeneity among studies on the association between spinopelvic alignment with PROMs such as variations in physical measures, PROMs, or follow-up timepoints, it was not possible to conduct a cross-sectional or longitudinal synthesis of the findings. Studies on LBP also could not be synthesized due to heterogeneity. Detailed results of individual studies are presented in S5 Appendix.”

Reviewers' comments: - Line 144: ChatGPT was used in the translation of the foreign language text. Although the translation probably works quite well, I’ m not sure if it has been validated as a proper translation of scientific texts.

Response to reviewers: Thank you for your thoughtful comment. Our decision to use ChatGPT was informed by the literature highlighting its superior ability to preserve stylistic features and complex linguistic structures compared to Google Translate, as well as its good performance in translating medical and technical literature. However, as noted in previous studies, this approach may have some limitations. To ensure accuracy, all translations in our study were reviewed by native speakers of the corresponding language. The following statement has been added to the manuscript:

- Line 154-158: “We selected ChatGPT over Google Translate for its better preservation of stylistic elements, contextual nuances, and strong performance in medical translations[44]. However, as noted in previous studies, this approach may have some limitations [44, 45]. To ensure accuracy, all translations in our study were reviewed by native speakers of the respective languages.”

Reviewers' comments: - Lines 154 and 163: “a third reviewer”. You mean fourth? There was a mention of three authors who extracted data and that there was a separate mediator.

Response to reviewers: We are grateful for your careful review and valuable suggestion. We had two reviewers act as the second reviewer. One reviewer (SV) conducted the process for half of the studies with RR and the other half with LF—thus, each study was assessed by two reviewers. The third person (AR) served as a mediator and was available to resolve any disagreements that arose. We have clarified this in the revised manuscript as follows:

- Line 165 and 166: “We had two reviewers act as the second reviewer. One reviewer (SV) conducted the process for half of the studies in collaboration with RR and the other half with LF.”

Reviewers' comments: - Line 176: reference to [49] Pais et al (2020). How is the text related to the reference?

Response to reviewers: Thank you for your detailed and helpful observations. In the study by Pais et al. (2020), the authors defined thresholds for rating the NIH Quality Assessment Tool as follows: 'good' for >80% of questions rated as 'yes,' 'fair' for 60–80%, and 'poor' for <60%. As we adopted an approach previously used by Pais et al. (2020), we cited their study accordingly.

We also added a reference to support the explanation of publication bias in the GRADE assessment for accuracy.

Reviewers' comments: - Tables 3 and 4 are clear.

Response to reviewers: Thank you very much for your insightful feedback.

Reviewers' comments: Results

- Lines 221-236. I think there is way too much text. I find it difficult to follow the narrative. A simple flow chart is enough.

Response to reviewers: We value your feedback and have addressed your points accordingly. For lines 221 to 232, a supplementary file was created to provide detailed information on eligibility confirmation, including full-text article retrieval, verification of THA indication, and requests for separate THA results. A correction was made in the retrieval summary, as one study had been counted twice in the full-text retrieval narrative.

Reviewers' comments: - Table 2. This table has a summary of every article included to the review including name, publication year, study objectives, number of patients and sex, health conditions physical outcome measures, PROMs, follow-up time and summary of results. While its impressive, I find it very difficult to read, mainly due the length of up to 40 pages and the extensive use of abbreviations (footnotes are 15 lines long). Overall, its impossible to form a coherent summary of the results. I think it would be more beneficial if this table was transferred to supplementary data or to the end of the article and this table greatly shortened to include only key results of the each stydy.

Response to reviewers: We appreciate your thoughtful feedback on Table 2. Since this table contains all the important information on the studies, we have included it as a supplementary file to enhance the readability of the main text.

Reviewers' comments: - Lines 209-310: Again, I don’t think it’ s necessary to report lower back pain separately.

Response to reviewers: Thank you for your feedback. Given the importance of LBP in this population, and the inconsistent findings in existing studies, we defined a secondary objective to provide an overview of the available evidence on the association between physical measures with LBP. This could help address current knowledge gaps

---

## [Decision Letter · Decision Letter 1]

26 Sep 2025

Dear Dr.  Vatandoost,

We look forward to receiving your revised manuscript.

Kind regards,

Masaya Anan, Ph.D.

Academic Editor

PLOS ONE

Journal Requirements:

Additional Editor Comments:

Very Concise Academic Editor Comments

Please address the following points before the manuscript can be considered for acceptance:

Required:

Revise the title to align with the actual evidence base, or clearly state early that evidence on spinopelvic alignment is very limited and that the main synthesis concerns physical functioning.

Acknowledge and clarify the inclusion of both prognostic and cross-sectional studies, and note the implications of this heterogeneity.

Recommended:

Expand discussion of the evidence gap on spinopelvic alignment.

Conduct a final check for clarity and language.

Reviewer's Responses to Questions

**Comments to the Author**

Reviewer #1: All comments have been addressed

Reviewer #3: (No Response)

2. Is the manuscript technically sound, and do the data support the conclusions?

Reviewer #1: Yes

Reviewer #3: Yes

3. Has the statistical analysis been performed appropriately and rigorously?

Reviewer #1: Yes

Reviewer #3: Yes

4. Have the authors made all data underlying the findings in their manuscript fully available?

Reviewer #1: Yes

Reviewer #3: Yes

5. Is the manuscript presented in an intelligible fashion and written in standard English?

Reviewer #1: Yes

Reviewer #3: Yes

Reviewer #1: The authors did a good job and met all of the reviewers' comments. They edited the manuscript and enhanced the scope of the review. I was delighted reading the current version of their manuscript.

Reviewer #3: Major Comments and Suggestions:

Mismatch Between Title, Aims, and Actual Content:

Comment: The title and one of the primary objectives prominently feature "spinopelvic alignment." However, only 6 out of the 51 included studies specifically investigated this parameter under the strict inclusion criteria (requiring measurement points on both the spine and pelvis). This creates a significant discrepancy between the reader's expectation, set by the title, and the actual content of the review, which is predominantly a synthesis of evidence on broader "physical functioning" measures (e.g., strength, gait speed).

Suggestion:

Primary Suggestion: Revise the title. It is highly recommended to change the title to more accurately reflect the core content.

Enhanced Clarification: If the current title is retained, a very clear and prominent statement must be added early in the Abstract and Introduction explicitly stating that the evidence regarding spinopelvic alignment is extremely limited based on the defined criteria, and that the main synthesis pertains to physical functioning measures. The explanations added in the introduction (Lines 73-77) and discussion (Lines 514-518) are necessary but may be insufficient to fully manage reader expectations set by the title.

Clarity on the Type of "Association" Assessed:

Comment: The manuscript combines results from prognostic studies (e.g., using preoperative measures to predict postoperative outcomes) and cross-sectional studies (assessing correlations at a single time point, e.g., postoperatively). These study designs address different scientific questions (prediction vs. concurrent relationship). Combining them in a narrative synthesis may contribute to heterogeneity and obscure clear interpretations.

Suggestion: It is important to explicitly address this in the Methods (e.g., in inclusion criteria) or Results section. A brief statement should clarify that the review included studies examining any temporal association (predictive, concurrent, etc.), and acknowledge that this methodological heterogeneity might be a factor in the inconsistent findings. When presenting results, consider grouping or clearly labeling studies based on the type of association they assessed to aid interpretation.

Strengthening the Discussion on Evidence Gaps:

Comment: The discussion correctly notes the scarcity of evidence on spinopelvic alignment. However, this "evidence gap" itself should be framed as a key finding of the review.

**Do you want your identity to be public for this peer review?** For information about this choice, including consent withdrawal, please see our Privacy Policy

Reviewer #1: No

Reviewer #3: No

---

## [Author Response · Author response to Decision Letter 2]

7 Nov 2025

Reviewers’ comments (Editor Comments):

Required:

Revise the title to align with the actual evidence base, or clearly state early that evidence on spinopelvic alignment is very limited and that the main synthesis concerns physical functioning.

Response to reviewers:

Thank you for your feedback. It is now clearly stated in the abstract (page 2) that evidence on spinopelvic alignment is very limited, with the synthesis focusing predominantly on physical measures of physical functioning.

In Table 1 (page 7), under the inclusion criteria, a justification is provided for narrowing the focus to spinopelvic parameters defined by one point on the spine and one on the pelvis. It has also been explained in the Discussion section (page 33).

Reviewers’ comments (Editor Comments):

Acknowledge and clarify the inclusion of both prognostic and cross-sectional studies, and note the implications of this heterogeneity.

Response to reviewers:

Thank you for your feedback. This has been addressed in the Design section (page 8) of the inclusion criteria (Table 1), where it is clarified that no restrictions were applied to the type of association, with examples provided in parentheses. Additionally, the following explanation has been added to the limitation section (page 36) to clarify that methodological heterogeneity may have contributed to the observed inconsistencies in the findings.

-Line 589-591: “Since no limitations were placed on the type of association (e.g., concurrent, predictive, responsiveness, etc.), this methodological heterogeneity may contribute to the observed inconsistencies in the findings.”

Reviewers’ comments (Editor Comments):

Recommended:

Expand discussion of the evidence gap on spinopelvic alignment.

Response to reviewers:

Thank you for your feedback. It has been clarified in the discussion (page 33) by revision of the explanation and adding the following explanation:

-Line 520-531: “Only six studies have evaluated this relationship. However, in the scoping review by Pourahmadi et al. [20] on spinopelvic alignment and LBP following THA, more studies were identified. This discrepancy may reflect differences in how spinopelvic parameters were defined. In this review, spinopelvic alignment was narrowly defined as involving one parameter on the spine and one on the pelvis, which led to the exclusion of studies assessing pelvic parameters alone. The objectives also differed. Pourahmadi et al. [20] aimed to provide an overview of spinopelvic alignment and LBP following THA, whereas this review specifically focused on studies evaluating the association between spinopelvic alignment and LBP following THA. The limited number of studies addressing association further highlights a gap in the existing literature. This evidence gap limits our ability to draw firm conclusions about the influence of spinopelvic alignment on PROMs and underscores the need for standardized measurement approaches and systematic evaluation to enable more robust synthesis and clinical interpretation.”

Reviewers’ comments (Editor Comments):

Conduct a final check for clarity and language.

Response to reviewers:

We are grateful for your valuable suggestion. The files have been checked for clarity and language.

In the PRISMA diagram, the number of unretrieved full texts and the conference abstracts of the already included articles were reported together; for clarity, they are now presented separately, and the corresponding files were edited accordingly (Figure 1, S3 Appendix, manuscript).

One full text provided by the author, which had been inadvertently omitted, was added to the diagram in S3 Appendix.

Some minor edits were made to the manuscript, S3 Appendix, S4 Appendix, and S5 Appendix for improved clarity and consistency.

Reviewers’ comments (Reviewer 1):

The authors did a good job and met all of the reviewers' comments. They edited the manuscript and enhanced the scope of the review. I was delighted reading the current version of their manuscript.

Response to reviewers:

Thank you for your feedback. We are glad to hear that the revisions meet your expectations.

Reviewers’ comments (Reviewer 3):

Reviewer #3: Major Comments and Suggestions:

Mismatch Between Title, Aims, and Actual Content:

Comment: The title and one of the primary objectives prominently feature "spinopelvic alignment." However, only 6 out of the 51 included studies specifically investigated this parameter under the strict inclusion criteria (requiring measurement points on both the spine and pelvis). This creates a significant discrepancy between the reader's expectation, set by the title, and the actual content of the review, which is predominantly a synthesis of evidence on broader "physical functioning" measures (e.g., strength, gait speed).

Suggestions:

Primary Suggestion: Revise the title. It is highly recommended to change the title to more accurately reflect the core content.

Enhanced Clarification: If the current title is retained, a very clear and prominent statement must be added early in the Abstract and Introduction explicitly stating that the evidence regarding spinopelvic alignment is extremely limited based on the defined criteria, and that the main synthesis pertains to physical functioning measures. The explanations added in the introduction (Lines 73-77) and discussion (Lines 514-518) are necessary but may be insufficient to fully manage reader expectations set by the title.

Response to reviewers:

Thank you for your feedback. We recognize that most of the included studies focused on the physical measures of physical functioning. We also acknowledge that we defined strict criteria for spinopelvic parameters, requiring inclusion of both spinal and pelvic regions to ensure a focused and consistent approach. The six studies included in the review represent all available evidence that met these criteria, highlighting a clear gap in existing literature.

Modifying the study title at this point would conflict with the established methodological framework and could affect the transparency of the review process as the research question was developed following a preliminary scoping search and implemented according to a predefined, published, and PROSPERO-registered protocol. To provide further clarity, the following sentence has been added to the Abstract (page 2).

-Line 38-40: “The evidence on spinopelvic alignment was extremely limited (6 studies), with the main synthesis focusing predominantly on physical measures of physical functioning.”

The following justification for including only parameters with one point on the spine and one point on the pelvis has also been provided in the inclusion criteria (page 7) in Table 1.

-Page 7- Table 1: “To accurately characterize spinopelvic alignment, parameters defined by one point on the spine and one on the pelvis were included. This approach ensures that the selected measures specifically reflect the interaction between spinal and pelvic regions, rather than isolated regional parameters.”

Given the expectations set by the title and the limited evidence on the association between spinopelvic alignment and PROMs, the Discussion (page 33) and conclusion section (page 37) have been expanded accordingly.

-Line 520-531: “Only six studies have evaluated this relationship. However, in the scoping review by Pourahmadi et al. [20] on spinopelvic alignment and LBP following THA, more studies were identified. This discrepancy may reflect differences in how spinopelvic parameters were defined. In this review, spinopelvic alignment was narrowly defined as involving one parameter on the spine and one on the pelvis, which led to the exclusion of studies assessing pelvic parameters alone. The objectives also differed. Pourahmadi et al. [20] aimed to provide an overview of spinopelvic alignment and LBP following THA, whereas this review specifically focused on studies evaluating the association between spinopelvic alignment and LBP following THA. The limited number of studies addressing association further highlights a gap in the existing literature. This evidence gap limits our ability to draw firm conclusions about the influence of spinopelvic alignment on PROMs and underscores the need for standardized measurement approaches and systematic evaluation to enable more robust synthesis and clinical interpretation.” -Line 609-612: “A key finding of this review is that the limited and heterogeneous nature of available studies on the association between spinopelvic alignment and PROMs prevented meaningful synthesis, highlighting a clear gap in existing literature.”

Reviewers’ comments (Reviewer 3):

Clarity on the Type of "Association" Assessed:

Comment: The manuscript combines results from prognostic studies (e.g., using preoperative measures to predict postoperative outcomes) and cross-sectional studies (assessing correlations at a single time point, e.g., postoperatively). These study designs address different scientific questions (prediction vs. concurrent relationship). Combining them in a narrative synthesis may contribute to heterogeneity and obscure clear interpretations.

Suggestion: It is important to explicitly address this in the Methods (e.g., in inclusion criteria) or Results section. A brief statement should clarify that the review included studies examining any temporal association (predictive, concurrent, etc.), and acknowledge that this methodological heterogeneity might be a factor in the inconsistent findings. When presenting results, consider grouping or clearly labeling studies based on the type of association they assessed to aid interpretation.

Response to reviewers:

We appreciate your thorough review. As indicated in the inclusion criteria (Table 1), no restrictions were applied to the type of association. To provide further clarity, the term “concurrent” has been added to the list of examples in parentheses within the design section (page 8) of the inclusion criteria (Table 1). Additionally, it has been clarified in the limitation section (page 36) that methodological heterogeneity may have contributed to the observed inconsistencies in the findings.

-Line 589-591: “Since no limitations were placed on the type of association (e.g., concurrent, predictive, responsiveness, etc.), this methodological heterogeneity may contribute to the observed inconsistencies in the findings.”

Regarding the synthesis, it has been divided into two parts, with the interpretation separated based on study design. The sections are titled: “Associations between measures in cross-sectional studies” and “Associations between measures in cohort longitudinal studies”.

Reviewers’ comments (Reviewer 3):

Strengthening the Discussion on Evidence Gaps:

Comment: The discussion correctly notes the scarcity of evidence on spinopelvic alignment. However, this "evidence gap" itself should be framed as a key finding of the review.

Response to reviewers: Your feedback is greatly appreciated. To better reflect this in the study, we have added to the Conclusion (page 37) that the limited studies on the association between spinopelvic alignment and PROMs prevented meaningful synthesis, underscoring a clear gap in the literature.

-Line 609-612: “A key finding of this review is that the limited and heterogeneous nature of available studies on the association between spinopelvic alignment and PROMs prevented meaningful synthesis, highlighting a clear gap in existing literature.”

---

## [Editor Report · Decision Letter 2]

10 Dec 2025

Association between physical measures of spinopelvic alignment and physical functioning with patient reported outcome measures after total hip arthroplasty: systematic review and narrative synthesis

PONE-D-25-12916R2

Dear Dr. Sima Vatandoost,

We’re pleased to inform you that your manuscript has been judged scientifically suitable for publication and will be formally accepted for publication once it meets all outstanding technical requirements.

Kind regards,

Michela Saracco

Academic Editor

PLOS One
---

## [Editor Report · Acceptance letter]

PONE-D-25-12916R2

PLOS One

Dear Dr. Vatandoost,

I'm pleased to inform you that your manuscript has been deemed suitable for publication in PLOS One. Congratulations! Your manuscript is now being handed over to our production team.

Kind regards,

on behalf of

Dr. Michela Saracco

Academic Editor

PLOS One